# Chondroprotective Actions of Selective COX-2 Inhibitors In Vivo: A Systematic Review

**DOI:** 10.3390/ijms21186962

**Published:** 2020-09-22

**Authors:** Ufuk Tan Timur, Marjolein M. J. Caron, Ralph M. Jeuken, Yvonne M. Bastiaansen-Jenniskens, Tim J. M. Welting, Lodewijk W. van Rhijn, Gerjo J. V. M. van Osch, Pieter J. Emans

**Affiliations:** 1Laboratory for Experimental Orthopedics, Department of Orthopaedic Surgery, Maastricht University Medical Centre+, 6229 HX Maastricht, The Netherlands; marjolein.caron@maastrichtuniversity.nl (M.M.J.C.); r.jeuken@maastrichtuniversity.nl (R.M.J.); t.welting@maastrichtuniversity.nl (T.J.M.W.); l.van.rhijn@mumc.nl (L.W.v.R.); p.emans@mumc.nl (P.J.E.); 2Department of Orthopaedics, Erasmus MC University Medical Center, Wytemaweg 80, 3015 CN Rotterdam, The Netherlands; y.bastiaansen@erasmusmc.nl (Y.M.B.-J.); g.vanosch@erasmusmc.nl (G.J.V.M.v.O.); 3Department of Otorhinolaryngology, Erasmus MC University Medical Center, Wytemaweg 80, 3015 CN Rotterdam, The Netherlands

**Keywords:** selective COX-2 inhibitors, intra-articular injection, knee osteoarthritis, DMOADs

## Abstract

Knee osteoarthritis (OA) is a condition mainly characterized by cartilage degradation. Currently, no effective treatment exists to slow down the progression of OA-related cartilage damage. Selective COX-2 inhibitors may, next to their pain killing properties, act chondroprotective in vivo. To determine whether the route of administration is important for the efficacy of the chondroprotective properties of selective COX-2 inhibitors, a systematic review was performed according to the PRISMA guidelines. Studies investigating OA-related cartilage damage of selective COX-2 inhibitors in vivo were included. Nine of the fourteen preclinical studies demonstrated chondroprotective effects of selective COX-2 inhibitors using systemic administration. Five clinical studies were included and, although in general non-randomized, failed to demonstrate chondroprotective actions of oral selective COX-2 inhibitors. All of the four preclinical studies using bolus intra-articular injections demonstrated chondroprotective actions, while one of the three preclinical studies using a slow release system demonstrated chondroprotective actions. Despite the limited evidence in clinical studies that have used the oral administration route, there seems to be a preclinical basis for considering selective COX-2 inhibitors as disease modifying osteoarthritis drugs when used intra-articularly. Intra-articularly injected selective COX-2 inhibitors may hold the potential to provide chondroprotective effects in vivo in clinical studies.

## 1. Introduction

Knee osteoarthritis (OA) is a condition that leads to pain and is mainly characterized by cartilage degradation [1]. Currently, drug treatments provide symptomatic pain relief but no effective treatment exists to slow down progression of OA-related cartilage damage [1]. Pain-killing drug therapies include non-steroidal anti-inflammatory drugs (NSAIDs). NSAIDs provide pain relief by blocking cyclooxygenase (COX)-dependent prostanoid synthesis. Prostanoids are an important family of signaling molecules present in synovial fluid [2]. At least two COX isoforms have been described, COX-1 and COX-2, the latter being considered as the inflammatory isoform [3]. Selective COX-2 inhibitors have been developed to specifically target the inflammatory COX-2 while circumventing inhibition of the COX-1 isoform. While selective COX-2 inhibitors may provide an effective means for pain relief, targeting the inflammatory COX-2 may be also a promising approach to inhibit cartilage degradation and thereby slow down knee OA progression [3]. This hypothesis is supported by the accumulating evidence showing that inflammation precedes OA disease progression [3,4].

From the anti-nerve growth factor clinical trials it became clear that next to a substantial improvement in pain, patients also displayed structural OA disease progression [5]. This emphasizes the importance of treatment strategies not only providing pain relief, which will lead to a vicious self-perpetuating cycle of joint overloading and OA progression, but also providing the ability for disease modification.

The actions of non-selective COX inhibitors on cartilage degradation in vitro and in vivo have been excellently reviewed in the past [6]. Moreover, ex vivo and in vivo actions of the selective COX-2 inhibitor celecoxib on cartilage degradation, synovial inflammation and osteoclast metabolism have also been reviewed and date back to 2011 [7]. Recently, an updated review of studies published until 2016 has been conducted on the OA disease-modifying actions of celecoxib on different OA tissues. This review confirms the contradictive reports regarding the chondroprotective actions of celecoxib both ex vivo and in vivo. Based on these reviews, it still remains obscure whether selective COX-2 inhibitors can be used in vivo to protect cartilage and slow down the progression of knee OA. One of the explanations of the contradictive reports found in the literature regarding the potential of selective COX-2 inhibitors may be related to the route of administration. Specifically, scarcely vascularized tissues such as cartilage and meniscus, which are important participants in knee OA, may be modified differentially after systemic oral treatment compared to intra-articular treatment [8,9]. To date, the role of the route of administration on the chondroprotective effects of selective COX-2 inhibitors remains unclear. Therefore, the objective of this review was to systematically review available literature on chondroprotective properties of selective COX-2 inhibitors in preclinical models for OA or in clinical OA studies depending on the route of administration.

## 2. Results

The search strategy yielded 25 articles for inclusion (Figure 1). We identified preclinical and clinical studies investigating chondroprotective actions of selective COX-2 inhibitors either via systemic administration or via intra-articular delivery. Using the SYRCLE’s risk of bias tool for preclinical studies, we observed that outcome measurements were in general blinded, but randomization sequences were not always reported increasing the risk for selection bias (Appendix A) [10]. For clinical studies, the Cochrane risk bias tool revealed that clinical studies were non-randomized, displaying a high risk of selection and performance bias (Appendix A) [11].

The following sections will discuss the main results of the included 25 studies and compare chondroprotective actions of selective COX-2 inhibitors when used systemically or intra-articularly.

### 2.1. Preclinical Studies: Oral and Intraperitoneal Administration

Fourteen of the included studies have investigated chondroprotective actions of selective COX-2 inhibitors after systemic administration (Table 1). Most studies focused on chondroprotective actions of the selective COX-2 inhibitor celecoxib [12,13,14,15,16,17,18,19,20,21], while other studies investigated chondroprotective actions of meloxicam [22,23] and etoricoxib [24,25]. Four studies that investigated chondroprotective actions of selective COX-2 inhibitors in surgically induced OA models failed to demonstrate chondroprotective actions when the drug was administered directly after OA induction [12,14,18,22]. On the other hand, Dai et al. demonstrated chondroprotective actions of celecoxib, as evidenced by an improved macroscopic and histological OARSI score, in a surgically induced pig OA model, in which treatment was started one week after surgery [20]. Further support of chondroprotective actions of selective COX-2 inhibitors comes from a study in a surgically induced rat OA model, where treatment with oral etoricoxib decreased OA-related cartilage damage as evidenced by an improved Pritzker score compared to controls [24]. Chondroprotective actions of celecoxib were also shown in a study in which the Achilles tendon transection rat knee OA model was used [15]. It was shown that compared to controls, chondrocyte apoptosis was lower after oral administration of celecoxib [15].

Next to surgically induced OA models, the monosodium iodoacetate (MIA) rat OA model [16,17,23], the collagenase OA model [13,19] and the spontaneous OA mouse model (STR/Ort) [21] were also used to investigate chondroprotective actions of selective COX-2 inhibitors celecoxib [13,16,17,19,21] and meloxicam [23]. In these studies, chondroprotective actions of selective COX-2 inhibitors were observed after oral administration as evidenced by improved histological scores compared to the controls.

While the aforementioned studies investigated chondroprotective actions of drugs administered orally, one study investigated chondroprotective actions of etoricoxib when administered intraperitoneally [25]. In this study, treatment with intraperitoneal etoricoxib injections was started two days after surgically inducing OA using the destabilization of the medial meniscus (DMM) mouse model [25] and histological analysis did not demonstrate a reduction in OA-related cartilage damage.

In conclusion, there is conflicting evidence regarding the chondroprotective actions of systemically administered selective COX-2 inhibitors since five studies did not show chondroprotective actions, while nine studies showed chondroprotective actions after systemic administration with selective COX-2 inhibitors.

### 2.2. Preclinical Studies: Intra-Articular Administration

Next to in vivo studies investigating chondroprotective actions of selective COX-2 inhibitors administered orally, seven studies focused on their chondroprotective actions when used intra-articularly (Table 2). These studies evaluated several selective COX-2 inhibitors such as celecoxib [26,27,28,29], parecoxib [30], meloxicam [31] and etoricoxib [32]. All studies used surgically induced OA models. Chondroprotective actions, as evidenced by improved histological scores, were shown in three studies after intra-articular administration with selective COX-2 inhibitors [26,30,31]. Four studies investigated chondroprotective actions of selective COX-2 inhibitors when incorporated in a drug delivery system (DDS). Dong et al. used celecoxib-loaded liposomes or liposomes loaded with a combination of celecoxib and hyaluronic acid to study OA-related cartilage damage in a surgically induced rabbit OA model [27]. No difference in histological scores was reported when rats received celecoxib-loaded liposomes, while liposomes loaded with both celecoxib and hyaluronic acid exerted a chondroprotective effect compared to control injections [27].

In an earlier study [28], our group has investigated chondroprotective effects of celecoxib-loaded polyesteramide (PEA) microspheres in a surgically induced rat OA model. While celecoxib-loaded microspheres reduced PGE_2_ as measured in homogenates from knees in this experiment, we could not find chondroprotective effects of celecoxib-loaded PEA microspheres as measured by the OARSI score [28]. In another study, Tellegen et al. [29] evaluated effects of the aforementioned celecoxib-loaded PEA microspheres on cartilage degeneration in a surgically induced rat OA model. Consistent with our study, no chondroprotective action of celecoxib-loaded PEA microspheres was observed. In contrast to the findings of the aforementioned two studies, one study investigating controlled release of etoricoxib demonstrated chondroprotective actions as evidenced by improved histological scores compared to controls [32].

In conclusion, when used intra-articularly, three studies report chondroprotective actions of selective COX-2 inhibitors. On the other hand, there is conflicting evidence regarding chondroprotective actions of selective COX-2 inhibitors incorporated in intra-articular drug delivery systems: two studies failed to demonstrate chondroprotective actions, one study shows chondroprotective actions only when celecoxib is combined with HA and one study demonstrated chondroprotective activity.

### 2.3. Clinical Studies

Four clinical studies investigated chondroprotective actions of selective COX-2 inhibitors via oral administration, while there were no studies investigating chondroprotective actions of selective COX-2 inhibitors after intra-articular administration (Table 3).

Tindall et al. [33] included patients diagnosed with knee OA in a prospective open-label trial. Patients with OA received celecoxib orally at two dosages throughout a 12-month study period. Long-term celecoxib treatment did not result in significant changes of joint space width (JSW) at 1 year after treatment as assessed radiographically. Moreover, when radiographic knee OA progression during the study was compared to radiographic knee OA progression prior to the start of the study, no significant differences were detected [33].

Sawitzke et al. [34] conducted a placebo-controlled study of patients with knee OA in which patients received placebo or celecoxib orally on a daily basis. No significant differences were found in JSW loss compared to placebo at 24 months, while Kellgren and Lawrence grade 2 knees showed a trend towards less JSW loss. In contrast to the findings of the aforementioned clinical studies, a study performed by de Boer et al. suggested that celecoxib might exert chondroprotective actions [35]. In this study, patients with end-stage knee OA were treated with orally supplemented celecoxib 28 days prior to total knee replacement surgery. Compared to patients that did not receive celecoxib, cartilage samples of patients that received celecoxib showed increased proteoglycan (PG) synthesis, decreased PG release and an increased PG content.

Raynould et al. conducted a study in which patients with knee OA were enrolled receiving celecoxib orally throughout a 12-month study period [36]. After correcting for potential confounders, celecoxib treatment did not show any protective effect on cartilage volume loss compared to a historical knee OA cohort loss, as assessed by quantitative MRI.

In conclusion, evidence of chondroprotective actions of selective COX-2 inhibitors in human patients is low, since three human in vivo studies failed to show chondroprotective effects of orally administered celecoxib, while only one study suggests that celecoxib may act chondroprotective in humans.

The selectivity of the COX-2 inhibitors that were used in the preclinical and clinical studies described in this review has been compared by Riendeau et al. [37]. The COX-2 selectivity may play a role in the chondroprotective outcome of a selective COX-2 inhibitor, but the number of studies described in this systematic review was too small to draw any conclusions. An overview of the half maximal inhibitory concentration (IC_50_) values for COX-1 and COX-2 of these selective COX-2 inhibitors together with the COX-1/COX-2 IC_50_ ratio is provided in Table 4.

## 3. Discussion

The main finding of this systematic review is that the administration route plays a major role in determining chondroprotective actions of selective COX-2 inhibitors. The intra-articular administration route may be promising, since studies using bolus intra-articular administration of selective COX-2 inhibitors show chondroprotective effects. On the other hand, conflicting evidence exists when selective COX-2 inhibitors are incorporated into a drug delivery system. While preclinical studies point out to a potential chondroprotective role of COX-2 inhibitors, clinical studies did not investigate the intra-articular administration route and failed to confirm chondroprotective actions of systemically administered selective COX-2 inhibitors.

Discrepancies in chondroprotective actions of selective COX-2 inhibitors observed in the preclinical studies may be related to the route of administration. While six of the fourteen studies using systemic administration failed to demonstrate chondroprotective actions of selective COX-2 inhibitors, all studies that applied intra-articular bolus injections demonstrated chondroprotective actions. Interestingly, all clinical studies included in this systematic review evaluated chondroprotective actions using the systemic administration route. Since these studies failed to demonstrate chondroprotective actions, it will be of interest to investigate the chondroprotective actions of selective COX-2 inhibitors using the intra-articular administration route for clinical studies. We did not encounter studies comparing chondroprotective effects of systemic versus intra-articularly administration of selective COX-2 inhibitors, but we believe this will be of interest to investigate in the future.

Improved chondroprotection of intra-articular injections with selective COX-2 inhibitor compared to a saline control condition may be related to increased bioavailability of the drug in the knee joint compared to systemic administrations. In addition, in a total joint disease such as knee OA [38,39,40,41], intra-articular administration of drugs may be more effective compared to systemic treatment due to the presence of scarcely vascularized tissues such as cartilage and meniscus [8]. Since COX-2 inhibitors can have an effect on all joint tissues, it is unclear whether the chondroprotective effect is a direct result of COX-2 inhibition in chondrocytes or an effect of COX-2 inhibition of other joint tissues.

Others and we failed to show a reduction of OA-related cartilage damage by celecoxib, when incorporated in an intra-articular drug delivery system [28,29]. It is a possibility that prolonged release of celecoxib may counteract potential chondroprotective effects due to increased loading of the affected joint indirectly caused by the analgesic effects of COX-2 inhibitors. These findings corroborate earlier observations in clinical studies, in which patients treated with anti-NGF demonstrated an increase in OA-related cartilage damage possibly due to joint overloading [5].

COX-2 inhibitors are expected to have anti-inflammatory effects by inhibiting the synthesis of prostanoids. Prostanoid subtypes are considered as inflammatory mediators [2] and are involved in cartilage degradation [3], but also in other pathophysiologic OA processes in different joint tissues such as synovial fibrosis and chondrocyte hypertrophy [42,43]. However, also anti-inflammatory actions of certain prostanoid subtypes have been shown [2], and therefore identifying downstream targets of the COX-2 pathway may further aid in anti-inflammatory treatment for knee OA. Inflammatory processes have been suggested to precede knee OA [4,44] and to be involved in structural disease progression [4]. A window of opportunity may exist, in which modulating the inflammatory status of the knee joint via intra-articular treatment with selective COX-2 inhibitors may lead to OA disease modification. The initiation of treatment may thus be important for the treatment outcome. The studies performed by Wen et al. [24] and Dai et al. [20] did not start treatment shortly after surgery and demonstrated chondroprotective actions of orally administered COX-2 inhibitors. It can be hypothesized that inhibiting inflammation directly after inducing a joint trauma may compromise cartilage regeneration, since inflammation is part of the early phases of natural tissue regeneration after a trauma [45]. This may explain why studies in surgical OA models fail to show chondroprotective effects when starting treatment directly after surgery. Moreover, the lack of a chondroprotective effect in clinical studies may be related to the stage of disease. Patients with knee OA are diagnosed in a stadium when the disease has progressed towards its end-stage [46], and it can be hypothesized that at this stage the disease is in an irreversible stadium where drug-based disease modification is not effective anymore. In addition, knee OA is a heterogeneous disease showing variability in the rate of disease progression in human subjects [47], while also the existence of distinct OA subtypes has been suggested [18,48], suggesting that patient-tailored drug treatment needs to be developed. Finally, the outcome measurements of clinical studies, such as joint space narrowing on conventional radiography, may not be sensitive enough. With the ongoing advancements in cartilage imaging [49], advanced techniques such as 7-tesla MRI imaging [49] may provide a more sensitive means to evaluate multiple outcome domains relevant to the clinical and pathophysiological aspects of disease modifying osteoarthritis drug (DMOAD)-mediated disease modification.

## 4. Materials and Methods

### 4.1. Search Strategy and Data Extraction

MEDLINE and EMBASE databases were systematically searched on all studies relating intra-articular or oral treatment of knee OA patients with selective COX-2 inhibitors. The search was conducted in May 2020 according to the search strategy and data collection guidelines of the preferred reporting items for systematic reviews and meta-analyses (PRISMA) statement. A manual search of the Cochrane library yielded no relevant articles. Upon reading the full-text key papers snowballing searching manually reference lists of the included articles was allowed. Disagreements between the reviewers were resolved by consensus.

After disregarding duplicates, the title and the abstract of articles were independently screened by two observers according to predefined criteria. The search query was as follows:

(“Osteoarthritis”[Mesh] OR OA OR osteoarthritis) AND (“Knee”[Mesh] OR Knee [tiab]) AND (((COX-2 [tiab] OR COX2 [tiab] OR Cyclooxygenase 2 [tiab] OR”Cyclooxygenase 2”[Mesh]) AND (inhibitor [tiab] or inhibition [tiab] or limitation [tiab] or limiting [tiab])) OR celecoxib [tiab] OR etoricoxib [tiab] OR rofecoxib [tiab] or valdecoxib [tiab] or lumiracoxib [tiab] or mavacoxib [tiab] or meloxicam [tiab] or VA441 [tiab] or MK-0966 [tiab] or gw406381 [tiab] or SC-58635 [tiab] or VA692 [tiab] or VA694 [tiab] or SC-236 [tiab]) AND ((oral [tiab] AND (suppletion [tiab] OR supplements [tiab] OR supplementation [tiab] OR treatment [tiab] OR ingestion [tiab] OR medication [tiab] OR tablets [tiab] OR intake [tiab] OR absorption [tiab])) OR treatment OR ((intraarticular [tiab] OR intra-articular [tiab]) AND (injection [tiab] OR therapy [tiab] OR supplementation [tiab] OR suppletion [tiab]))).

### 4.2. Articles Were Selected Based on Inclusion and Exclusion Criteria

Inclusion criteria:Presenting data about chondroprotective effects of COX-2 inhibitors;The chondroprotective effect is defined as any effect that leads to significantly less cartilage degradation evidenced either through imaging, biochemical analysis or on histology.Either in vivo animal studies or clinical studies involving human knee OA patients;Intra-articular therapies with COX-2 inhibitors;Systemic therapies with COX-2 inhibitors.

Exclusion criteria:
Studies solely investigating non-selective cox-inhibitors;Studies investigating actions of celecoxib inhibitors on pain modulation in knee OA;Studies investigating other joints than the knee;Studies reporting solely in vitro data.

Two observers systematically extracted study data based on the inclusion and exclusion criteria. The risk of bias of the included studies was evaluated using the SYRCLE’s risk of bias tool for animal studies [10] and the Cochrane risk of bias tool [11] for human studies.

## 5. Conclusions

To date there is conflicting evidence regarding the ability of selective COX-2 inhibitors to be used as DMOADs. Preclinical studies have used different routes of administration, which may alter the chondroprotective outcome of selective COX-2 inhibitors in the knee joint where scarcely vascularized tissues are present. Other factors such as the OA model type, type of selective COX-2 inhibitor and disease stage seem also to be involved in the chondroprotective outcome of selective COX-2 inhibitors. Despite the limited evidence of data in clinical studies, there seems to be a preclinical basis for considering selective COX-2 inhibitors as DMOADs specifically when used intra-articularly.

## 6. Patents

TJM Welting is listed as inventor on patents: WO2017178251, WO2017178253 and US 20130123314. PJ Emans and LW van Rhijn are listed as inventors on patent US 20130123314. LW van Rhijn, PJ Emans and TJM Welting have shares in Chondropeptix and are CDO, CMO and CSO of Chondropeptix, respectively.

## Figures and Tables

**Figure 1 ijms-21-06962-f001:**
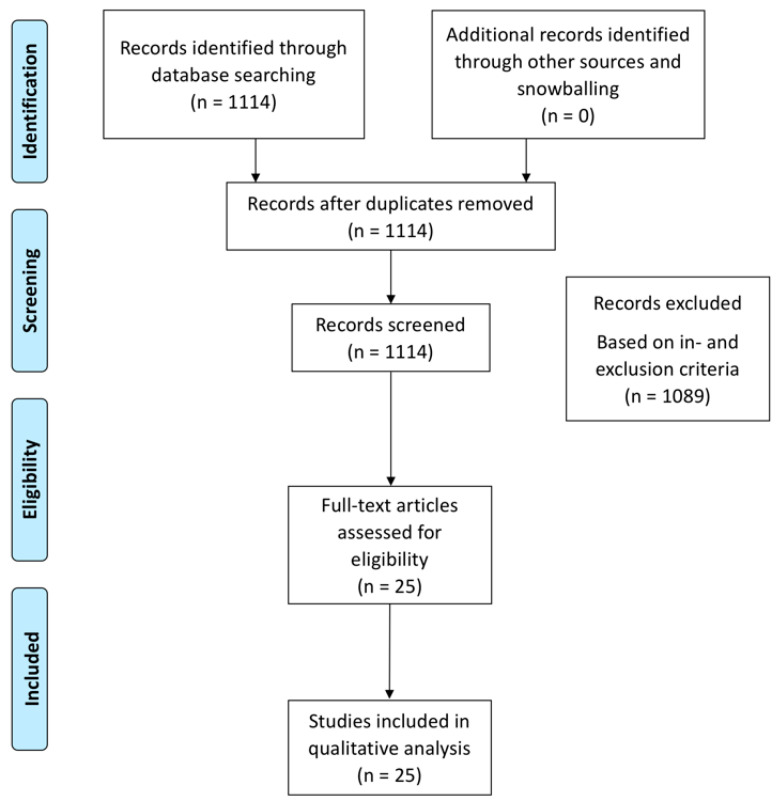
PRISMA flowchart showing the yield of the search and inclusion of studies leading to the 25 included studies.

**Table 1 ijms-21-06962-t001:** The schematic of studies investigating chondroprotective effects of systemically administered selective COX-2 inhibitors.

**Preclinical Studies Using Oral Administration**								
**Authors**	**Species**	**COX-2 Inhibitor**	**OA Model**	**Dosing Regime**	**Start of Treatment**	**Timepoint of Evaluation**	**Evaluation of Cartilage Degradation**	**Main Findings**
Mastbergen et al. [12]	Canine	Celecoxib	Groove	Daily 200 mg	Directly after surgery	15 weeks after OA induction	Histology:Modified Mankin Biochemistry: PG content, synthesis and release	No difference in histological scores, no difference in PG content, synthesis or release
Huh et al. [13]	Rabbit	Celecoxib	Collagenase	Daily 100 mg/kg	Directly after OA induction	4 weeks after OA induction	Histology: Colombo score for cartilage and synovitis score for synoviummRNA analysis *PTGS1, PTGS2*, *MMP1* and *MMP3*	Improved histological score in celecoxib group vs. control, decreased MMP-1 mRNA expression in celecoxib group vs. control
Jones et al. [22]	Rat	Meloxicam	MCLT + ACLT + MMx	Daily 3 mg/kg	Directly after OA induction	8 weeks after OA induction	Histology:Modified Mankin	No difference in histological score meloxicam vs. control
Fukai et al. [14]	Mouse	Celecoxib	MCLT + MMx	Daily 10 mg/kg or 30 mg/ kg	Directly after surgery	12 weeks after OA induction	Histology:Pritzker score	No difference in Pritzker score celecoxib vs. control.
Ou et al. [15]	Rat	Celecoxib	Achilles tendon transection	Daily 24 mg/kg	Directly after surgery		Histology:Type II collagen Tunel staining	No difference in type II collagen content in celecoxib vs. control, decreased chondrocyte apoptosis in celecoxib group.
Ashkavand et al. [16]	Rat	Celecoxib	MIA	Daily 100 mg/kg	Directly after OA induction	15 days after OA induction	Histology: Own developed score	Improved histological score in the celecoxib group. Superior chondroprotective effects when celecoxib is combined with silymarin
Moon et al. [17]	Rat	Celecoxib	MIA	Daily 2.5 mg/kg	Directly after OA induction	7 days after OA induction	Histology:Modified Mankin	No difference in Modified Mankin celecoxib vs. control. Synergistic beneficial action when celecoxib is added to rebamipide
Panahifar et al. [18]	Rat	Celecoxib	ACLT + MCLT + MMx	Daily 2.86 mg/kg	Directly after surgery	4,8 and 12 weeks after OA induction	Histology: Modified Mankin	No difference in histological score celecoxib vs. control.
Li [19]	Rat	Celecoxib	Collagenase	Daily 0.25 mg	6 weeks after surgery	10 weeks after OA induction	Histology: Colombo score and Biochemistry:CTX-II content in serum, Caspase 3 activity in tissue homogenate	Improved histological score in celecoxib group vs. control, higher CTX-II content in celecoxib group, lower Caspase 3 activity in celecoxib group. More pronounced effects when celecoxib is combined with diacerein
Dai [20]	Pig	Celecoxib	MMx	20 mg/kg daily	1 week after surgery	12 weeks after surgery	Macroscopic:|OARSI scoreHistology:OARSI scoreCOL-II and AGC immunohistochemistry	Improved macroscopic and histological score celecoxib group vs. control, no difference in COL-II and AGC expression
Nagy [23]	Rat	Meloxicam	MIA	Daily 0.2 mg/kg or 1 mg/kg	3 weeks after OA induction	11 weeks after OA induction	Histology: OARSI score	Improved histological score at both doses meloxicam compared to control
Tu [21]	Mouse	Celecoxib	Spontaneous OA (STR/Ort mouse)	Daily 8 mg/kg	3 months old mice	4 weeks after start treatment	Histology: OARSI score	Improved OARSI score in celecoxib group vs. controls,
Wen [24]	Rat	Etoricoxib	ACLT	6.7 mg/kg or 33.3 mg/kg three times per week	8 weeks after surgery	21 weeks after surgery	Histology: Pritzker score for cartilage	Improved histological score in the etoricoxib group versus control
**Preclinical Studies Using Intra-Peritoneal Injections**								
Liu [25]	Mouse	Etoricoxib	DMM	5 mg/kg, 10 mg/kg, 20 mg/kg three times per week	2 days after OA induction	30 days after surgery	Histology: OARSI score	No difference in histological score etoricoxib versus control.

ACLT: Anterior Cruciate Ligament Transection, CLX: celecoxib, DMM: destabilization medial meniscus, ETX: etoricoxib, MCLT: Medial Collateral Ligament Transection, MIAl: Monosodium Iodoacetate, MMx: medial meniscectomy, NP: nanoparticle, pMMx: partial medial meniscectomy, PG: proteoglycan.

**Table 2 ijms-21-06962-t002:** The schematic of studies investigating chondroprotective effects of intra-articularly administered selective COX-2 inhibitors.

	Preclinical Studies Using Intra-Articular Injections							
Authors	Species	COX-2 Inhibitor	OA Model	Dosing Regime	Start of Treatment	Timepoint of Evaluation	Evaluation of Cartilage Damage	Main Findings
Jean et al. [30]	Rat	Parecoxib	ACLT	Weekly 100 µg parecoxib for 5 consecutive weeks	Eight weeks after surgery	20 weeks after surgery	Histology: Mankin score	Improved histological scores in the parecoxib group compared to controls
Jiang et al. [26]	Rabbit	Celecoxib	ACLT+ PCLT + MMx	Weekly 1.2 mg celecoxib for 5 consecutive weeks	Directly after surgery	12 weeks after surgery	Histology: Mankin score	Improved histological scores in the Celecoxib group compared to controls
Dong et al. [27]	Rat	Celecoxib	ACLT+ PCLT+ MCLT + MMx	Single injection: 0.15 mg celecoxib incorporated in DDS	One week after surgery	2 weeks after surgery	Histology: Colombo score	Improved histological score only when celecoxib is combined with HA in a DDS compared to controls
Wen et al. [31]	Rat	Meloxicam	ACLT	Weekly 0.25 or 1 mg meloxicam for 5 consecutive weeks	Five weeks after surgery	20 weeks after surgery	Histology: Pritzker score	Lower Pritzker score in the meloxicam group versus control
Janssen et al. [28]	Rat	Celecoxib	ACLT + pMMx	Single injection: 0.015 mg celecoxib incorporated in DDS	Four weeks after surgery	16 weeks after surgery	Histology: OARSI score	No difference in histological score in celecoxib loaded microspheres vs. control
Tellegen et al. [29]	Rat	Celecoxib	ACLT + pMMx	Single injection: 0.015, 0.115 or 0.195 mg celecoxib incorporated in DDS	Four weeks after surgery	16 weeks after surgery	Histology: OARSI score	No difference in histological score in celecoxib loaded microspheres vs. control
Liu et al. [32]	Rat	Etoricoxib	ACLT	Three injections: - 10 µM etoricoxib in 100 µL NaCl- 6.93 μg ETX-NPs, drug loading unclear	Three, six and nine weeks after surgery	12 weeks after surgery	Histology: OARSI score Immunohistochemistry	Improved histological score in ETX-NP, but not ETX, compared to control.

ACLT: Anterior Cruciate Ligament Transection, CLX: celecoxib, DDS: drug delivery system, ETX: etoricoxib, HA: hyaluronic acid, MCLT: Medial Collateral Ligament Transection, MMS: modified mankin score, MMx: medial meniscectomy, NP: nanoparticle, pMMx: partial medial meniscectomy.

**Table 3 ijms-21-06962-t003:** The schematic overview of clinical studies investigating chondroprotective effects of systemically administered selective COX-2 inhibitors.

Clinical Studies Using Oral Administration							
Authors	Species	COX-2 Inhibitor	OA Grade	Dosing Regime	Treatment Duration	Evaluation of Cartilage Damage	Chondroprotection
Tindall et al. [33]	Human	Celecoxib	K&L 2 and 3	Daily 200 or 400 mg	12 months	Radiographs of the knee	No difference in JSW, subchondral sclerosis, cysts, malalignment or tilting after 12 months of treatment with celecoxib
Sawitzke et al. [34]	Human	Celecoxib	K&L 2 and 3	Daily 200 mg	24 months	Radiographs of the knee	No significant differences in JSW loss in celecoxib vs. controls
De Boer et al. [35]	Human	Celecoxib	Not specified	Daily 200 mg	4 weeks	Biochemical cartilage analysis:PG synthesis rate, PG release and PG content	Increased PG synthesis rate, decreased PG release and increased PG content in the celecoxib group vs. controls. Decreased release of il-1, TNF-α and mmp-activity in celecoxib group vs. control
Raynauld et al. [36]	Human	Celecoxib	K&L 2 and 3	Daily 200 mg	12 months	Quantitative MRI	No difference in cartilage volume loss in celecoxib group vs. a historical cohort control group

JSW: Joint space width, K&L: Kellgren and Lawrence, PG: proteoglycan.

**Table 4 ijms-21-06962-t004:** The schematic overview of the half maximal inhibitory concentration (IC_50_) values and the COX-1/COX-2 IC_50_ ratio of various selective COX-2 inhibitors. Adapted from [37].

Drug	IC50 COX-1 (µM)	IC50 COX-2 (µM)	COX-1/COX-2 IC_50_ Ratio
Meloxicam	1.4	0.7	2
Celecoxib	6.7	0.87	7.7
Etoricoxib	116	1.1	105
Parecoxib	26.1	0.87	30

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
