# Peer review of "Chondroprotective Actions of Selective COX-2 Inhibitors In Vivo: A Systematic Review"

_ijms, 2020, doi:10.3390/ijms21186962_

Round 1

Reviewer 1 Report

ijms-931722

Chondroprotective actions of selective COX-2 inhibitors in vivo: a systematic review

Timur et al.

This review summarizes pre-clinical and clinical studies on chondroprotective effects of selective COX-2 inhibitors applied by different route of administration. Results of pre-clinical studies showed that intra-articular injection of selective COX-2 inhibitors had advantage in relation to systemic administration and slow releasing system.

Minor comments:

  1. The table showing different COX-2 selective inhibitors mentioned in the review, their mode of action and targets are recommended.
  2. The role of anti-inflammatory prostaglandins must be mentioned and discussed.

Author Response

Dear reviewer, 

Thank you for your constructive feedback on our initial submission.

We have carefully considered your comments and suggestions. Please find below a point by point response on your comments:

This review summarizes pre-clinical and clinical studies on chondroprotective effects of selective COX-2 inhibitors applied by different route of administration. Results of pre-clinical studies showed that intra-articular injection of selective COX-2 inhibitors had advantage in relation to systemic administration and slow releasing system.

Minor comments:

1. The table showing different COX-2 selective inhibitors mentioned in the review, their mode of action and targets are recommended.

We have now included a new table describing the COX-2 selective inhibitors mentioned in the review. Moreover the table demonstrates the half inhibitory concentration of various selective COX-2 inhibitors for COX-1 and COX-1 and provide a COX-1/COX-2 IC50 ratio (lines 185-190, table 4). 

2. The role of anti-inflammatory prostaglandins must be mentioned and discussed.

As suggested a sentence mentioning the anti-inflammatory actions of certain prostanoid subtypes is mentioned in the discussion (line 232-234).

Reviewer 2 Report

Interestin argument, well described analysis and considerations

I found the argument of this manuscript very interesting and quite original, as in the majority of studies regarding COX 2-inhibitors , the efficacy of the drug is valued on the basis of pain relief, while the real problem of the patient is the loss of cartilage, leading to stronger shear stress and thus to pain. The conclusion of the paper may suggest that the next field of research should regard the reduction of arthritis , rather than of pain.

Author Response

Dear reviewer, 

Thank you for your generous comments on our manuscript and the critical assessment of our work. We are grateful for your positive evaluation of our work.